# Factors Predicting Post-Traumatic Positive and Negative Psychological Changes Experienced by Nurses during a Pandemic COVID-19: A Cross-Sectional Study

**DOI:** 10.3390/ijerph19127073

**Published:** 2022-06-09

**Authors:** Grzegorz Józef Nowicki, Barbara Ślusarska, Bożena Zboina, Aneta Jędrzejewska, Marzena Kotus

**Affiliations:** 1Department of Family and Geriatric Nursing, Medical University of Lublin, Staszica 6 Str., PL-20-081 Lublin, Poland; basiaslusarska@gmail.com (B.Ś.); jedrzejewska.ab@gmail.com (A.J.); 2Department of Pedagogy and Health Sciences, College of Business and Entreprise, Akademicka 20 Str., PL-27-400 Ostrowiec Świetokrzyski, Poland; bozenazboina@poczta.fm; 3Department of Anaesthesiological and Intensive Care Nursing, Medical University of Lublin, Chodźki 7 Str., PL-20-093 Lublin, Poland; marzenakotus@umlub.pl

**Keywords:** COVID-19, pandemic, nurses, changes in outlook, post-traumatic growth, influencing factors

## Abstract

It is common knowledge that COVID-19 affects physiopathological changes in all systems of the human body. On the other hand, events related to the COVID-19 pandemic also have a significant impact on the social and mental sphere of human functioning. The aim of this study is to determine the relationship between selected sociodemographic variables and selected subjective cognitive resources, and the positive and negative perception of the consequences of the COVID-19 pandemic in a group of nurses working in Poland. The computer-assisted web interviewing method was conducted between 1 and 15 May 2020. Participants were requested to complete the following questionnaires: The Changes in Outlook Questionnaire (CIOQ), The Impact Event Scale-Revised (IES-R), The Multidimensional Scale of Perceived Social Support (MSPSS), The Safety Experience Questionnaire (SEQ), and The Meaning in Life Questionnaire (MLQ). Three-hundred and twenty fivenurses working all over Poland participated in the study. Their mean age was 39.18 ± 11.16 years. A higher average level was noted among the surveyed nurses in the Positive Change subscale (18.56 ± 4.04). In a multivariate model, taking into account both sociodemographic and cognitive variables, the level of perceived traumatic stress, the level of social support, a sense of security, reflection on safety and a sense of meaning and meaning in life were independent predictors of a positive perception of the consequences of the COVID-19 pandemic. Those variables explained as much as 37% of the dependent variable, and the nature of the relationship was positive. While we are still a long way from understanding the full range of the long-term impact of the COVID-19 pandemic on mental health and psychosocial well-being, it is possible that in this challenging context there are many individual resources available to perceive the effects of the current pandemic positively. Therefore, they should be strengthened through the development and implementation of intervention programs to improve the mental state of nurses.

## 1. Introduction

According to the World Health Organization (WHO) [1], the Corona Virus Disease 2019 (COVID-19) pandemic of March 2020 [2] is the most significant global public health threat, placing enormous pressure on healthcare systems. Among the various occupational groups, health care workers (HCW) are most vulnerable to adverse effects related to the pandemic, especially in the field of mental health [3]. Moreover, HCWs represent systemically important professions, which means that their work is necessary, and the demand for it increases in a pandemic.

COVID-19 influences physiopathological changes in all human body systems, especially the immune and respiratory systems. On the other hand, events related to the pandemic also significantly impact the social and mental sphere of human functioning [4,5]. Research shows that HCWs are at high risk of developing mental stress and other adverse mental health symptoms due to exposure to COVID-19 in healthcare settings [6,7]. The results of the available literature reviews also emphasise that fear of infection and transmission of the virus to family members, colleagues and friends is the dominant concern among HCW, limiting their social interactions and the most important risk factor for their physical functioning, as well as well-being and mental health [8]. In addition, there are well-founded concerns about the mental health, mental adjustment, and recovery of HCW caring for COVID-19 patients during and after the pandemic.

Previous studies conducted during epidemics such as the Severe Acute Respiratory Syndrome (SARS) of the 2003 Middle East Respiratory Syndrome (MERS) epidemic of 2013–2016 and Ebola of 2014–2016 indicate adverse psychological effects connected to those epidemics, such as anxiety, depression among HCWs, burnout and post-traumatic stress disorder (PTSD), and their symptoms persist from one to three years [9,10,11]. On the other hand, a meta-analysis concerning the psychological impact of the COVID-19 pandemic on HCWs showed that the total incidence of depression was 21.7%, anxiety was 22.1%, and PTSD was 21.5% [12]. Another meta-analysis showed that among HCWs, the highest incidence of PTSD was recorded among nurses, followed by physicians and other medical professionals (physiotherapists, medical caregivers) [13]. Furthermore, a metanalysis by Vizheh et al. [14] showed that the mental burden affected 12% to 67% of nurses during the COVID-19 pandemic. In contrast, the meta-analysis by Ślusarska et al. [15] showed that during the COVID-19 pandemic, the prevalence of depression among nurses was 22%, while anxiety disorders were experienced by 29% of them.

Even though a traumatic event can cause post-traumatic symptoms (PTS), it can also become a catalyst for positive change. This phenomenon is called post-traumatic growth (PTG). According to the concept of Calhoun and Tedeschi [16], PTG defines a positive psychological change that follows very stressful and demanding life situations [17]. This means that people have the ability to grow despite experiencing trauma. HCWs have great potential to develop PTG due to their personal and professional characteristics. Olson et al. [18] and Huecker et al. [19] emphasise the great importance of the study of PTG and its predictors among HCW during the COVID-19 pandemic. Despite the high interest of researchers in the negative mental health consequences of the COVID-19 pandemic, research has begun documenting the positive psychological effects of the pandemic. For example, moderately increased levels of PTG have been found in frontline nurses and have been associated with social support [20]. The narrative review showed that HCW working on the frontline had a higher level of PTG compared to HCW away from the frontline [21]. In contrast, studies by Vazquez et al. [22] conducted amongst the general adult population in Spain showed that PTG was associated with primal beliefs about a good world, openness to the future and identification with humanity.

Gathered and analysed research indicates that nurses are the group most at risk of developing PTSD during the COVID-19 pandemic. Therefore, examining the positive and negative perceptions of the COVID-19 pandemic and the predictive factors in this professional group seems to be quite significant. The main mechanisms responsible for the negative consequences of the COVID-19 pandemic may be primarily related to the level of anxiety and depression. Negative feelings can worsen the mental and physical well-being of HCW, undermining their confidence and sense of security and even meaning in life. Personal resources usually play an essential role in risk perception during a pandemic, with its negative consequences, as well as in shaping the well-being and positive consequences of COVID-19 stress in HCW. Personal resources are generally regarded as qualities that are valued by a person and are able to improve their effective functioning in terms of control and impact on the environment [23]. In creative adaptation, subjective personal resources can be used to assist individuals in the effective reinterpretation of challenging and stressful life experiences and ineffective adaptation to professional tasks. The risk of developing COVID-19 increases the occupational requirements of HCWs, leading to a decline in well-being with negative consequences. At the same time, perceptions of social support, security and meaning in life can mitigate the adverse effects and develop positive growth associated with the pandemic.

Studies have shown that meaning in life was positively associated with well-being among American HCW [24], but negatively associated with depression in Turkish HCW [25]. In studies in the Israeli adult population conducted during the COVID-19 pandemic, perceived support from a loved one negatively correlated with the severity of depression, anxiety, obsessive-compulsive disorder (OCD) and PTSD [26]. In contrast, a qualitative study on the protection of HCW against exposure to SARS-CoV-2 during patient care during a pandemic found that the sense of security in terms of confidence in personal protective equipment (PPE) and infection prevention and control (IPC) strategies was associated with a lower level of emotional exhaustion [27]. Given that the COVID-19 pandemic is a prolonged stressful situation, especially for HCWs, the availability of social support, a sense of security, and finding meaning and meaning in life would enable workers to cope with stress and promote well-being and positive functioning.

Therefore, considering the need to understand these phenomena, the aim of this study was to determine the relationship between selected sociodemographic variables and selected subjective cognitive resources, and the positive and negative perception of the consequences of the COVID-19 pandemic in the group of nurses working in Poland. The second goal of the research was to determine which sociodemographic variables and cognitive factors explain the variability of positive and negative perceptions of the consequences of the COVID-19 pandemic. We are particularly interested in factors with a buffer effect for the positive aspects of nurses’ perceptions of the aftermath of a pandemic, such as perceived social support, experiencing safety, and a sense of meaning and meaning in life.

## 2. Materials and Methods

### 2.1. Study Design and Participants

The analysis was conducted using data collected as part of a cross-sectional study between 1 and 15 May 2020. The detailed study protocol in terms of study stages, participant inclusion and exclusion criteria and data collection methods is described elsewhere [28]. A summary of this analysis is provided below. The research material was collected among 325 nurses using the computer-assisted web interviewing (CAWI) method due to the restrictions related to the COVID-19 pandemic in terms of social isolation.The questionnaire was posted on the “Google Surveys” portal and the link to the questionnaire was promoted twice: on the first and the seventh day of the study on the ten most popular fan pages addressed to nurses on Facebook. Participants were allowed to complete the survey only once.The completion of online questionnaires is an established method in healthcare research [29].

After giving informed consent to participate in the study, in order to verify the respondents, the information that the study was granted to nurses and the question, “Are you a nurse?” appeared on the next page. The respondent could respond “Yes” or “No”. In the case of marking the answer “No”, the questionnaire was closed automatically, thanking them for their time. To be eligible to participate in the study, respondents had to meet certain inclusion and exclusion criteria. The inclusion criteria included: (1) a nurse working during the COVID-19 epidemic, defined as the period from 20 March 2020; (2) professional activity before the coronavirus epidemic, meaning during January and February 2020; and (3) informed consent to participate in the study by responding “Yes”. The exclusion criteria were: (1) being on sick leave, maternity, parental or care leave prior to the announcement of the epidemic in Poland (January and February 2020); (2) being on sick leave, maternity, educational or care leave after the announcement of the epidemic in Poland; (3) withdrawal from work for health reasons; and (4) refusal to give informed consent to participate in the study. There is no target recruitment size. As direct comparisons are not being drawn, a power calculation has not been performed.

### 2.2. Study Questionnaire

In order to achieve the objectives of the study, a questionnaire consisting of five standardized tools and an original tool were used. All respondents completed the same questionnaire. In the instructional manual concerning each listed tool, the respondents were asked to rate certain factors, taking into account the current epidemiological situation. A detailed description of the questionnaires used is presented in another publication [28], while a short description of the scales is provided below. All the scales used in the study were characterised by optimal internal consistency, presented in our earlier publication in Appendix A [28]. The standardised research scales used in the study include:

-Changes in Outlook Questionnaire (CIOQ). The scale was developed by Joseph et al. [30]. At the same time, in our research, we used the scale in the Polish adaptation of Skalski [31], and it contains 10 statements, five each for two subscales: Positive Change and Negative Change of consequences related to a traumatic event.-Impact Event Scale-Revised (IES-R). The scale was developed by Weiss and Marmara [32] in the Polish adaptation of Juczyński and Ogińska-Bulik [33] to assess traumatic stress, including disturbing memories and persistent negative emotions related to trauma. In the analysis of the obtained results, we adopted the approach that the diagnosis of PTSD can be suspected only in those people who score above the cross-over point (>1.5) in the overall score and in each of the three dimensions.- Multidimensional Scale of Perceived Social Support (MSPSS). The scale was developed by Zimeta et al. [34] Inthe Polish adaptation of Buszman and Przybyły-Basista, [35] assesses the perception of social support taking into account three primary sources of support: significant others, family, and friends.-Safety Experience Questionnaire (SEQ). The scale by Klamut [36] assesses the level of experiencing security. The scale is an operationalization of a two-factor model, in which two subscales have been distinguished: a sense of safety (the level of safety experience related to the current satisfaction of basic needs, having satisfactory living conditions and the ability to act) and reflection on safety (the degree of considering matters related to their own safety, their loved ones’ safety, and the safety of the nation and the world in the assessment of life situations and social reality).-Meaning in Life Questionnaire (MQL). The scale was developed by Steger et al. [37] and the tool used in the research was adapted by Kossakowska et al. [38]. The questionnaire consists of 10 questions and examines two dimensions: presence and search.

The questionnaire was supplemented with a personal information form in order to collect information on several basic sociodemographic data: gender, age, marital status, place of residence, the respondent’s cohabitants, whether they have children, education, completed postgraduate education, seniority as a nurse, position held, whether the respondent took care of a suspected or diagnosed patient with SARS-CoV-2 at work, and whether the respondent participated in training on the use of personal protective equipment, and the functioning of the medical facility where they worked during the COVID-19 pandemic.

### 2.3. Ethical Considerations

Ethical approval was issued by the Bioethics Committee at the Medical University of Lublin (decision number: KE-0254/73/2020). The research was conducted in accordance with the ethical principles contained in Recommendations from the Association of Internet Researchers [39]. Participation in the study was voluntary and anonymous. All study participants gave their informed consent to participate in the study electronically. The informed consent form preceding the questionnaire contained an explanation of the purpose, subject of the research, the approximate duration of the study and the method of answering the questionnaire. After reading the information about the survey, the respondent was asked to express their willingness to participate in the survey by clicking “Yes” or withdrawing from the survey by closing the page in the web browser containing the survey or selecting the “No” option. Only those who chose “Yes” were transferred to the questionnaire page. The respondent could resign from the survey at any time by closing the website with the questionnaire. We have described a detailed method of obtaining informed consent elsewhere [28].

### 2.4. Statistical Analysis

Continuous variables were presented as means (M) with standard deviation (SD). The Shapiro-Wilk test was used to assess conformity with a normal distribution. Categorical variables were reported as absolute numbers and percentages. Differences between groups were assessed by *t*-test or analysis of variance (ANOVA). Pearson correlation was used to investigate the relationships between numerical variables. Simple and multiple linear regression models were performed to assess the significant predictors of CIOQ—Positive Change or Negative Change. The variables with *p*-value < 0.1 were included in the multiple regression model. Three sets of models were constructed: Model A was performed for each independent variable separately (univariable analysis); Model B, included sociodemographic or cognitive factors, which were significant in the simple model; Model Cincluded all significant variables in simple models. The coefficient of determination (R^2^) was provided to describe the adequateness of fit for the performed models. Moreover, in the case of IES-R, MSPSS and MLQ scales, a strong correlation between subscales was observed. One of the subscales—Total Score (which was the strongest related to CIOQ)—was used for each scale to avoid collinearity in multivariable models. Additionally, the analysis restricted to women was performed.Statistical analyses were conducted using IBM Corp. (released in 2017) and IBM SPSS Statistics for Windows, Version 25.0. (IBM Corp, Armonk, NY, USA). Statistical A *p*-value < 0.05 was considered significant for two-tailed tests.

## 3. Results

### 3.1. Characteristics of Participants

Three hundred twenty-five nurses working all over Poland participated in the study, most of whom were women (96.7%, *n* = 311). The mean age was 39.18 ± 11.16 years. Most of the respondents lived in a city (66.46%, *n* = 216). 57.75% (*n* = 188) of the respondents were married, while the rest were single, widowed, or divorced. 67.7% (*n* = 220) of the respondents lived with their family, 20.3% with a partner, 3.4% (*n* = 11) with a roommate, and 8.6% (*n* = 28) lived alone. 65.2% (*n* = 212) had children living in their house. 43.7% (*n* = 142) had a master’s or higher education (more than a master’s). Postgraduate education finished by the surveyed nurses: 44.6% (*n* = 145) completed the specialization training, and 42.2% (*n* = 137) completed a qualification course in the field of nursing. 76% (*n* = 247) of respondents were nurses employed in the ward, 13.6% (*n* = 44) were a two-ward nurse, 4.9% (*n* = 13.6%) were a head nurse, and 5.5% (*n* = 18) were a primary care nurse. 46.5% (*n* = 151) cared for a patient with COVID-19, and 67.4% (*n* = 219) received training concerning the use of personal protective equipment and the functioning of the medical facility in which they work during the COVID-19 pandemic.

### 3.2. Distribution of the Analysed Features According to Scales CIOQ, IES-R, MSPSS, SEQ and MLQ

Table 1 presents the results of the respondents on the scales used in the study. Among the surveyed nurses, a higher average level was noticed in the subscale of perception of positive consequences of the COVID-19 pandemic (18.56 ± 4.04).

### 3.3. Relationship between Selected Sociodemographic Variables and the Assessment of Positive and Negative Consequences of the COVID-19 Pandemic

Table 2 presents the relationships between sociodemographic variables and positive and negative consequences related to the traumatic event, which is the COVID-19 pandemic. Age was negatively related to the Positive Change subscale (r = −0.15, *p* = 0.007), but no significant relationship was observed between age and the Negative Change subscale (r = 0.04, *p* = 0.48). Respondents living in cities (18.55 ± 4.06) and being widowed or divorced (17.91 ± 3.08) obtained a significantly lower mean value of the Positive Change subscale compared to respondents from rural areas (19.49 ± 3.94) and those who were married (19.41 ± 3.87) or living alone (18.19 ± 4.45). The other analysed variables did not significantly differentiate the mean scores on the Positive Change subscale in the study group.

In the case of the Negative Change subscale, education, completed postgraduate education, marital status and having children, significantly differentiated mean values were obtained. Higher mean values of this subscale occurred among respondents with a master’s degree or higher education, respondents who completed aqualification course in postgraduate education, and those living alone and without children.

### 3.4. Relationship between Positive and Negative Consequences Related to the Traumatic Event Which Is the COVID-19 Pandemic and Selected Cognitive Factors

Table 3 shows the relationship between the CIOQ subscales and selected cognitive variables. Significant and positive relationships were observed between the CIOQ Positive Change subscale and the level of perceived traumatic stress, and reflection on safety, with the strongest relationship between the sense of meaning and meaning in life (r = 0.403, *p* < 0.001) and the sense of social support (r = 0.401, *p* < 0.001). Conversely, there was no significant correlation between the CIOQ Positive Change subscale and the sense of security.

In the case of the second CIOQ Negative Change subscale, it was observed that the level of social support, the sense of security and the sense of meaning and meaning in life were negatively correlated with this subscale, while the level of perceived traumatic stress was positively related. The strongest correlation occurred in the case of experienced traumatic stress (r = 0.481, *p* < 0.001). There was no significant correlation between the CIOQ Negative Change subscale and reflection on safety.

### 3.5. Features Related to the Positive and Negative Perspective of the COVID-19 Pandemic—Multivariable Analysis

Table 4 and Table 5 present univariable models (Models A) and multivariable models (Models B) presenting the results of analysis of features significantly related in one-dimensional models for sociodemographic features and cognitive factors, (Model C) taking into account both sociodemographic features and cognitive determinants. In the case of the CIOQ Positive Change subscale, among the analysed sociodemographic features, only marital status was a significant predictor, with age explaining only 3% of the variability of this variable. The cognitive determinants which significantly related to the CIOQ Positive Change subscale included the level of perceived traumatic stress, the level of social support, a sense of security, reflection on safety, and a sense of meaning and meaning in life (Model B), explaining in total 36% of the variability of the dependent variable. In the full multivariable model (Model C), none of the analysed sociodemographic features turned out to be an independent predictor of the CIOQ Positive Change subscale value, and all of the analysed cognitive features explained as much as 37% of the dependent variable, and the nature of the relationship was positive.

In the case of the CIOQ Negative Change subscale, in the multivariable model, among the analysed sociodemographic features, marital status, education and completion of a qualification course in postgraduate education were significantly related to this variable and explained 8% of its variability (Model B). In Model B (among cognitive traits), the level of perceived traumatic stress, the level of social support and the sense of meaning and meaning in life were significantly related to the CIOQ Negative Change subscale, explaining in total 30% of its variability. In the full multivariable model (Model C), variables such as education, completion of a postgraduate qualification course, level of perceived traumatic stress, level of social support, and sense of meaning and meaning in life turned out to be independent predictors of the CIOQ Negative Change subscale (Model C). Together, these features accounted for 38% of the variability of the dependent variable. However, the relationship of such traits as the sense of social support, the sense of security and the sense of meaning and meaning in life in relation to the CIOQ Negative Change variable was negative.

The results of the analysis restricted to female participants are presented in Appendix A, Appendix A. Features significant in the analyses conducted on the entire sample (both including women and men) maintained statistical significance as well as the direction and strength of dependence among women.

## 4. Discussion

The COVID-19 pandemic has severe, multi-faceted consequences for people’s psychosocial and mental health [40], especially HCW. Therefore, a better understanding of the underlying protective factors and risks of both negative and positive psychological effects of a pandemic [41] is warranted. Given the need to investigate both the protective factors and risk factors associated with the negative and positive psychological consequences of the current global COVID-19 pandemic, the presented research aimed to identify the relationship between selected socio-demographic variables and cognitive factors, and positive and negative perceptions of the COVID-19 pandemic in the aforementioned group of nurses. Overall, the results of our study indicate that the sociodemographic variables significantly differentiating the surveyed group of nurses in terms of the perception of positive and negative consequences of the COVID-19 pandemic were age, place of residence, education, postgraduate education, marital status, and having children. However, when considering the analysed cognitive variables, it transpired that the majority was associated with a positive and negative perception of the consequences of the COVID-19 pandemic. Although the strongest positive perception of the consequences of the COVID-19 pandemic was associated with the sense of meaning and meaning in life and the perception of social support, the level of perceived traumatic stress was most strongly associated with the negative perception of the consequences of the COVID-19 pandemic. In multivariate models, the analysed sociodemographic and cognitive variables explained 37% of the variable of positive perception of the consequences of the COVID-19 pandemic and 38% of the variable of the negative perception of the consequences of the COVID-19 pandemic, of which, in the analysis of the positive perception of the consequences of the COVID-19 pandemic, none of the analysed sociodemographic features proved to be an independent predictor.

Moreover, the positive perception of the consequences of the COVID-19 pandemic has decreased. Additionally, it was observed that nurses living in rural areas and those who were married were characterised by a higher positive perception of the consequences of the COVID-19 pandemic, whereas a significantly lower level of perception concerning the negative consequences of the COVID-19 pandemic was observed in respondents characterised as having lower education, nurses who completed a qualifying nursing course, and those who were married and those who had children. Cui et al. [42] conducted a study among 167 frontline nurses during the COVID-19 pandemic in China, in Henan and Hubei provinces. In this research, PTG was measured using the Post-traumatic Growth Inventory (PTGI). Their results indicated that, as in our study, married nurses had higher PTG levels.

Conversely, the results of the cited studies indicate that, unlike the authors’ research, senior nurses and nurses with a higher level of education had a higher PTG score, while having children did not differentiate the group. Peng et al. [20] conducted a study amongst 116 frontline nurses during the COVID-19 pandemic, where PTG was significantly higher in nurses with children. At the same time, such variables as age, marital status, and education did not differentiate the study group. There are several mechanisms explaining the influence of some sociodemographic variables on the level of PTG. One of the sociodemographic variables which can increase PTG is having children. It relates to the duties and role of the mother, making the woman bolder and stronger in the face of new difficulties and challenges. Psychological research on frontline nurses has shown that identifying with the role of “mother” influences the level of PTG [43]. Updegraff and Taylor [44] proposed an explanation of the impact of being married on a higher level of PTG by linking it with positive mental development after trauma through a support system provided by another close person. However, our research only observed that the respondents remaining in a relationship had a significantly lower level of negative perception of the COVID-19 pandemic. Another variable is the age of the respondent, which in the case of nurses is associated with longer work experience, therefore some studies indicate that senior nurses had a significantly higher PTG [42] This may be related to the fact that nurses with morework experience show higher levels of critical thinking [45], but the results of our research have not confirmed this. A similar mechanism may apply to the level of education and PTG.

The second group of analysed variables that may influence the positive and negative perceptions of the COVID-19 pandemic were cognitive variables that can be involved in creative adaptation to help individuals effectively reinterpret difficult and stressful situations. In our research, the level of traumatic stress was weakly correlated and positively correlated with the positive perception of the consequences of the COVID-19 pandemic, and moderately correlated and positively correlated with the negative perception of the consequences of the COVID-19 pandemic. It should be emphasised that in both analysed situations, the correlations are positive, although the higher strength of the correlation of traumatic stress associated with COVID-19 was associated with a negative perception of the consequences of this pandemic. This discovery was partially in line with our expectations, because the PTG theory was proposed as a possible positive psychological consequence of the encountered traumatic events [46,47], and the individual’s perception of a traumatic event becomes a necessary condition for development [48,49]. On the basis of the obtained research results, it can be hypothesized that too high an intensity of traumatic stress causes a negative perception of the consequences of a pandemic. Chen et al. [50] obtained similar results amongst nurses, while Park et al. [51] confirmed this relationship among Amazon MTurk employees. Thus, PTG is associated with the symptoms of post-traumatic stress disorder and can be treated as a coping mechanism in the face of persistent suffering from trauma [52].

Tedeschi and Calhoun [53] stressed the great importance of social support as a direct predictor of PTG. People who experienced a traumatic event with a high level of social support more often received emotional or material support from family members, friends, or various social groups [54]. Conversely, one of the ways to combat the spread of the SARS-CoV-2 virus was the introduction of social isolation rules, limiting social support through face-to-face contact. This was especially truein the case of nurses, who were more exposed to the virus and became more isolated than the rest of society due to the nature of their work. The authors’ results showed that the sense of social support was positively correlated with a positive and negatively correlated with a negative perception of the COVID-19 pandemic. Obtained results were confirmed in previous studies among nurses during the COVID-19 pandemic [55,56,57]. One possible explanation is that high perceived social support can provide a sense of having a safe environment, emphasize feelings of belonging, serve as a buffer against stress, provide new meaning, and generate more positive perceptions, which endorse growth [58].

Safety is one of the most important categories which allows a description of the context of human life and the way people function [59]. In the context of experiencing security, the emotional aspect is feeling, while the rational, cognitive aspect is a reflection on security [60]. During the COVID-19 pandemic, especially at its beginning, nurses faced numerous problems that disturbed their sense of security and influenced their reflection on safety. These problems include, among others, a shortage of personal protective equipment, nursing staff shortages, fear for the health of oneself and relatives, etc. The authors’ research revealed the sense of security and reflection on safety as positively correlated with a positive perception of the COVID-19 pandemic, while the feeling of safety negatively correlated with a negative perception of the COVID-19 pandemic. Other research has confirmed that fear of infection and awareness of the risks are associated with PTG, but the relationship between the availability of personal protective equipment and PTG has not been confirmed [42,61]. Unfortunately, there are no studies assessing the impact of the sense of security on PTG among health care workers on the level of PTG. Therefore, further research is required to assess this aspect considering many intermediary variables, such as the availability of personal protective equipment, the workplace, the level of knowledge about the virus and many others.

According to the literature, the confrontation of an individual with stressful life events accompanied by various losses poses a challenge to the desire to perceive the world as meaningful and predictable, and thus may contribute to a search for meaning [62]. Earlier studies have shown a positive relationship between the presence of meaning in life and PTG [63,64]. Our research confirmed this fact, which showed a positive relationship between the search for meaning in life and a positive and negative relationship between the negative perception of the consequences of the COVID-19 pandemic. Unfortunately, there are few studies assessing the impact of meaning in life on PTG during the COVID-19 pandemic. Interesting observations about the sense of meaning and importance of life and other elements of psychological functioning during the COVID-19 pandemic were made by Baños et al. [65]. They assessed, inter alia, the meaning in life in various periods of the COVID-19 pandemic and found that its level was stable over time in the assessed periods of the lockdown. On the other hand, Trzebiński et al. [66] showed that the level of meaning and meaning in life positively correlates with lower anxiety and lower stressrelated to COVID-19. The explanatory mechanism may be that the search for meaning and meaning in life allows the individual to positively reassess traumatic events, strengthen the psychological resources needed to rediscover themselves, restore the individual to a basic, complex world, and be oriented towards future goals [67].

### 4.1. Implications for Nursing Practice and Education

The practical implications of our research indicate the importance of social support, a sense of security, reflection on safety and a sense of the meaning and meaning of life as protective factors in creating post-traumatic positive psychological changes in the face of this and future pandemics. Accordingly, we recommend that healthcare leaders provide support from their supervisors and develop safe practice procedures. Highly engaged and participatory leadership facilitates dealing with group problems, sharing and processing ideas, and empathetic team leaders can provide an understanding of nurses*’* needs and awareness of the thoughts and feelings of the nursing staff [68]. Another aspect of the practical implications in creating post-traumatic positive psychological changes should include conducting psychological interventions in nurses working in environments with high stress related to contact with an infected patient. Both psychologists, as well as direct close associates, can provide psychological first aid [69].

As for the educational aspects, they apply not only to the nurses themselves, working in direct contact with the infected patient but also to healthcare leaders in understanding the problem and organizational support in solving new problems that arise in the face of a pandemic. Our research indicates the critical role of cognitive variables in mitigating the adverse psychological effects of a pandemic. Therefore, it is vital to develop cognitive resources during both undergraduate and postgraduate education to be easier for nurses, with the participation of nursing leaders, to cope with the adverse psychological consequences of a pandemic. Nurses who can cope well with the adverse psychological effects of a pandemic will contribute to helping individuals in society cope with stress when an epidemiological emergency occurs. This research can contribute to the development of the nursing practice by identifying some significant cognitive resources contributing to the protection of the mental health of oneself, one*’*s family and the community, which can be developed during training on how to deal with a pandemic.

### 4.2. The Strengths and Limitations

The strengths and weaknesses of this study deserve consideration. Firstly, to our knowledge, it is one of the few studies which assesses, inter alia, the impact of the sense of security, reflection on safety and the meaning and meaning of life on the positive and negative perception of the consequences of the COVID-19 pandemic carried out with a group of nurses. Secondly, in our research we used standardised questionnaires which had good reliability coefficients and were adapted to the assessment for the COVID-19 pandemic [28]. Thirdly, the nurses we studied worked in Poland, where they had not participated in combating the COVID-19 pandemic for many decades.

Our research has several limitations. First, the study’s cross-sectional design is a constraint from which we can infer correlation, not construct causation. Our sample of nurses is not a nationwide representative sample. The average age of the respondents was lower than the average age of nurses in Poland. In our sample, the mean age was 39.18 ± 11.16, while the mean age of nurses in Poland, according to the statistics of the Supreme Chamber of Nurses and Midwives, in 2020, was 53.16 years [70]. The age difference between the group we studied and nurses in general, is particularly visible in the overrepresentation of the youngest age group of 21–30 years.

Nevertheless, the research results may be helpful in modeling research hypotheses and recommendations in the implementation of psychological care during a pandemic, especially among the younger age group of nurses. It is worth emphasizing that the study was conducted at the beginning of the COVID-19 pandemic in conditions of epidemic restrictions. The only chance to obtain data was to conduct CAWI studies, which was associated with more frequent participation in the study by younger people. Secondly, we adopted an online dissemination strategy due to the limitations of social contacts. Therefore, it was not possible to collect data on people who refused to participate in the study, and no percentage of refusals was recorded. In addition, when recruiting participants in the study, we relied on access to social networking sites, which is why the surveyed population does not include participants who do not have access to social networking sites and nurses who do not use these sites. Third, the study was conducted at the beginning of the COVID-19 pandemic, and as some authors suggest, PTG measured shortly after contact with a traumatic event may be an initial coping strategy [22]. Nevertheless, the results of our study can provide a valuable reference point for the discussion of PTG studies conducted in later or post-pandemic periods. Fourth, the research subject was Polish nurses, whose cultural context may cause their responses to differ from those of nurses working in other countries. Fifthly, filling in the questionnaire on their own could cause the respondents to underestimate or exaggerate the severity of certain symptoms to minimize or exacerbate their problems.

## 5. Conclusions

The COVID-19 pandemic quickly altered the working conditions for nurses in Poland, increasing the level of traumatic stress related to the pandemic. The authors’ observations revealed the prevalence of positive post-traumatic psychological changes among nurses. It has been observed that younger nurses living in rural areas and are married more often perceive the consequences of a pandemic positively. Whereas respondents with a higher education, with a qualification in the field of nursing, single people and those without children perceive the negative effects of the COVID-19 pandemic significantly more often. In a multivariate analysis, lower levels of perceived traumatic stress and higher levels of social support, a sense of security, reflection on safety, and a sense of meaning and meaning in life are the main sources of a 37% buffering explaining the variability of the prospect of positive consequences of the COVID-19 pandemic among nurses. Factors predicting post-traumatic negative psychological changes experienced by nurses during the COVID-19 pandemic in a multidimensional model reveal 38% of variability of negative consequences, and, among them, apart from sociodemographic features, a high level of perceived traumatic stress and low levels of social support, sense of security and sense and importance are important.

Our research clarifies the insufficient knowledge concerning the predictors of post-traumatic positive and negative psychological changes experienced by nurses during the COVID-19 pandemic. Moreover, we were able to identify the importance of the level of perceived traumatic stress, social support, sense of security, reflection on safety, and the sense of meaning and importance of life as protective factors in the mechanisms of creating post-traumatic positive psychological changes.

## Figures and Tables

**Table 1 ijerph-19-07073-t001:** Distribution of the analysed features in scales.

Scales	M ± SD
CIOQ—Positive Change	18.56 ± 4.04
CIOQ—Negative Change	14.28 ± 4.49
IES-R—Total score	1.78 ± 0.65
MSPSS—Total score	65.9 ± 13.3
SEQ—Sense of safety	3.23 ± 0.79
SEQ—Reflection on safety	4.21 ± 0.49
MLQ—Total score	5.33 ± 0.87

M: mean; SD: standard deviation; CIOQ: Changes in Outlook Questionnaire; IES-R: Impact Event Scale-Revised; MSPSS: Multidimensional Scale of Perceived Social Support; SEQ: Safety Experience Questionnaire; MLQ: Meaning in Life Questionnaire.

**Table 2 ijerph-19-07073-t002:** Associations between selected sociodemographic variables on the assessment of positive and negative consequences of the COVID-19 pandemic.

Variable	CIOQ—Positive Change		*p*	CIOQ—Negative Change		*p*
M	SD	M		SD		
Age (year)	r = −0.15		0.007	r = 0.04		0.48
Place of residence:						
Urban area	18.55	4.06	0.046	14.12	4.60	0.368
Rural area	19.49	3.94		14.60	4.28	
Education:						
Bachelor’s degree	18.87	3.86	0.99	13.54	4.30	0.008
Master’s degree or above	18.87	4.27		14.85	4.30	
Postgraduate education:						
Postgraduatediploma	19.25	4.30	0.42	4.47	14.00	<0.001
Qualificationcourse	19.09	3.82		15.42	4.50	
Specialisttrainingcourse	18.54	4.16		13.28	4.27	
Marital status:						
Married	19.41	3.87	0.016	13.67	3.96	0.006
Single	18.19	4.45		15.42	5.27	
Divorced/Separated/Widowed	17.91	3.08		14.12	4.14	
Living arrangements:						
Family	19.11	4.0	0.28	14.17	4.55	0.76
Cohabitant/Flat mate/Roommate	18.36	4.31		14.40	4.41	
Alone	18.32	3.47		14.78	4.37	
Child(ren) in House:						
No	18.44	4.53	0.166	15.04	5.40	0.025
Yes	19.09	3.74		13.87	3.88	
Rotating shift schedule:						
No	18.47	4.26	0.234	13.91	4.65	0.316
Yes	19.05	3.93		14.45	4.42	
Have you nursed a patient diagnosed with COVID-19:						
No	18.69	3.69	0.41	14.09	3.92	0.405
Yes	19.07	4.41		14.50	5.07	
Was there any training related to the coronavirus epidemic at work:						
No	18.41	4.53	0.16	14.70	4.54	0.244
Yes	19.09	3.76		14.08	4.47	

M: mean; SD: standard deviation; CIOQ: Changes in Outlook Questionnaire.

**Table 3 ijerph-19-07073-t003:** The relationship between the CIOQ subscales and selected cognitive variables.

Variable		CIOQ—Positive Change	CIOQ—Negative Change
IES-R—Total score	r	0.147	0.481
	*p*	0.008	<0.001
MSPSS—Total score	r	0.401	−0.205
	*p*	<0.001	<0.001
SEQ—Sense of safety	r	0.298	−0.307
	*p*	0.298	<0.001
SEQ—Reflection on safety	r	0.386	0.021
	*p*	<0.001	0.704
MLQ—Total score	r	0.403	−0.269
	*p*	<0.001	<0.001

CIOQ: Changes in Outlook Questionnaire; IES-R: Impact Event Scale-Revised; MSPSS: Multidimensional Scale of Perceived Social Support; SEQ: Safety Experience Questionnaire; MLQ: Meaning in Life Questionnaire; r: correlation coefficient.

**Table 4 ijerph-19-07073-t004:** Relationship between the positive outlook of the COVID-19 pandemic and selected sociodemographic and cognitive variables.

Variables	Changes in Outlook Questionnaire—Subscale Positive Change										
	Model A			Model B				Model C			
Sociodemographic variables:	b	SE	*p*	b	SE	*p*	R^2^	b	SE	*p*	R^2^
Age	0.014	0.02	0.483				3%				37%
Place of residence (reference category: Urban area)	0	0	0	0	0	0		0	0	0	
Rurar area	0.944	0.472	0.046	0.678	0.481	0.160		0.612	0.393	0.121	
Education (reference category: Bachelor’s degree)	0	0	0	0	0	0		0	0	0	
Master’s degree or above	−0.003	0.452	0.995								
Postgraduate education (reference category: Specialist training course)	0	0	0	0	0	0		0	0	0	
Postgraduate diploma	0.711	0.701	0.311								
Qualification course	0.543	0.481	0.260								
Marital status (reference category: Married)	0	0	0	0	0	0		0	0	0	
Single	−1.217	0.489	0.013	−1.072	0.499	0.032		−0.532	0.411	0.197	
Divorced/Separated/Widowed	−1.500	0.755	0.048	−1.356	0.760	0.076		−0.254	0.626	0.685	
Living arrangements (reference category: Family)	0	0	0	0	0	0		0	0	0	
Cohabitant/Flat mate or Roommate	−0.750	0.534	0.161								
Alone	−0.792	0.809	0.328								
Child(ren) in House (reference category: No)	0	0	0	0	0	0		0	0	0	
Yes	0.652	0.469	0.166								
Rotating shift schedule (reference category: No)	0	0	0	0	0	0		0	0	0	
Yes	0.574	0.481	0.234								
Have you nursed a patient diagnosed with COVID-19 (reference category: No)	0	0	0	0	0	0		0	0	0	
Yes	0.371	0.449	0.410								
Was there any training related to the coronavirus epidemic at work? (reference category: No)	0	0	0	0	0	0		0	0	0	
Yes	0.672	0.477	0.160								
Cognitive variables:											
IES-R—Total score	0.917	0.342	0.008	1.756	0.309	<0.001	36%	1.690	0.311	<0.001	
MSPSS—Total score	0.122	0.015	<0.001	0.056	0.016	<0.001		0.059	0.016	<0.001	
SEQ—Sense of safety	1.154	0.262	<0.001	1.354	0.262	<0.001		1.329	0.263	<0.001	
SEQ—Reflection on safety	2.801	0.424	<0.001	1.165	0.424	0.006		1.125	0.424	0.008	
MLQ—Total score	1.868	0.236	<0.001	1.261	0.236	<0.001		1.201	0.237	<0.001	

Model A: univariable analysis; Model B: included significant factors in univariable analysis (performed separately for sociodemographic and cognitive factors); Model C: included all significant factors in univariable analysis; CIOQ: Changes in Outlook Questionnaire; IES-R: Impact Event Scale-Revised; MSPSS: Multidimensional Scale of Perceived Social Support; SEQ: Safety Experience Questionnaire; MLQ: Meaning in Life Questionnaire; b: standardised beta coefficient; SE: standard error.

**Table 5 ijerph-19-07073-t005:** Relationship between the negative perspective of the COVID-19 pandemic and selected sociodemographic and cognitive variables.

Variables	Changes in Outlook Questionnaire—Subscale Negative Change										
	Model A			Model B				Model C			
Sociodemographic variables:	b	SE	*p*	b	SE	*p*	R^2^	b	SE	p	R^2^
Age	−0.061	0.022	0.007	−0.019	0.033	0.560	8%	−0.031	0.028	0.26	38%
Place of residence (reference category: Urban area)	0	0	0	0	0	0		0	0	0	
Rular area	0.476	0.528	0.368								
Education (reference category: Bachelor’s degree)	0	0	0	0	0	0		0	0	0	
Master’s degree or above	−1.310	0.498	0.009	−1.148	0.493	0.02		−0.977	0.410	0.018	
Postgraduate education (reference category: Specialist training course)	0	0	0	0	0	0		0	0	0	
Postgraduate diploma	0.717	0.763	0.348	0.512	0.764	0.503		0.543	0.644	0.400	
Qualification course	2.141	0.523	<0.001	1.592	0.587	0.007		1.305	0.491	0.008	
Marital status (reference category: Married)	0	0	0	0	0	0		0	0	0	
Single	1.748	0.542	0.001	1.626	0.941	0.085		1.558	0.788	0.049	
Divorced/Separated/Widowed	0.446	0.837	0.595	0.802	0.833	0.336		1.097	0.700	0.118	
Living arrangements (reference category: Family)	0	0	0	0	0	0		0	0	0	
Cohabitant/Flat mate or Roommate	0.230	0.596	0.700								
Alone	0.613	0.904	0.498								
Child(ren) in House (reference category: No)	0	0	0	0	0	0		0	0	0	
Yes	−1.172	0.520	0.025	0.954	0.957	0.320		0.755	0.799	0.345	
Rotating shift schedule (reference category: No)	0	0	0	0	0	0		0	0	0	
Yes	0.538	0.536	0.316								
Have you nursed a patient diagnosed with COVID-19 (reference category: No)	0	0	0	0	0	0		0	0	0	
Yes	0.417	0.500	0.405								
Was there any training related to the coronavirus epidemic at work? (reference category: No)	0	0	0	0	0	0		0	0	0	
Yes	−0.620	0.531	0.244								
Cognitive variables:											
IES-R—Total score	3.332	0.338	<0.001	2.722	0.358	<0.001	30%	2.757	0.348	<0.001	
MSPSS—Total	−0.069	0.018	<0.001	−0.042	0.018	0.022		−0.05	0.018	0.005	
SEQ—Sense of safety	−1.863	0.308	<0.001	−0.678	0.303	0.026		−0.663	0.294	0.025	
SEQ—Reflection on safety	0.844	0.499	0.092	1.149	0.492	0.02		1.160	0.479	0.016	
MLQ—Total score	−1.388	0.276	<0.001	−0.956	0.274	0.001		−0.783	0.265	0.003	

Model A: univariable analysis; Model B: included significant factors in univariable analysis (performed separately for sociodemographic and cognitive factors); Model C: included all significant factors in univariable analysis; CIOQ: Changes in Outlook Questionnaire; IES-R: Impact Event Scale-Revised; MSPSS: Multidimensional Scale of Perceived Social Support; SEQ: Safety Experience Questionnaire; MLQ: Meaning in Life Questionnaire; b: standardised beta coefficient; SE: standard error.

## Data Availability

The datasets used and/or analyzed during the current study areavailable from the corresponding author on reasonable request.

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
