# Peer review of "Factors Predicting Post-Traumatic Positive and Negative Psychological Changes Experienced by Nurses during a Pandemic COVID-19: A Cross-Sectional Study"

_ijerph, 2022, doi:10.3390/ijerph19127073_

Round 1

Reviewer 1 Report

Thank you very much for the opportunity to review this study. I consider the topic original and relevant. I have only a few recommendations:
1.    The abstract is too extended; it would be helpful to shorten it.
2.    Although the authors refer to the description of the research method in another text, it would be helpful to include the primary data from the detailed study protocol briefly.
3.    It is not clear whether all measurement instruments were distributed simultaneously in one data collection and to the same respondents; please explain in more detail
4.    The authors admit that the research population is not representative in the study's limitations. However, it would be helpful to describe whether the research at least attempted to make the sample representative or to what extent the sample differs from the population. Given the aim (to look for a link with socio-demographic factors), this question is crucial. The relevance of the results would need to be discussed more widely
5.    Are the authors considering repeating the study to compare any attitude changes?
6.    Are there any practical implications from the study, for example, in nurse education?

Reviewer 2 Report

This study analysed the background factors related to the psychological burden on a certain number of nurses in the early stages of the Covid-19 pandemic by means of a questionnaire, and the analysis methods, etc., were sound and the results seem to be valid.

However, it cannot be denied that the study is a little late in coming, as a considerable number of similar studies have been reported so far.

In addition, the results are enumerative because they are not in the form of a test of hypotheses using a model behind them, and the results do not seem to directly suggest a useful policy of care of nursing workers under COVID-19 pandemic.

Men were included in 14 out of 325 (3.3%), but they are too small, only work to add any extra disturbance to the results, and the analysis should have been restricted to women only from the outset.

The use of structural analysis of covariates might produce results that are easier to interpret, although it might be difficult to identify causal relationships a priori.

Anyway, the paper has been properly analysed and submitted in a sound form, so it is considered to be of value as a source of useful information.

Reviewer 3 Report

Dear Author:

This study aims to determine the relationship between selected sociodemographic variables and selected subjective cognitive resources and the positive and negative perceptions of COVID-19 pandemic consequences in a group of nurses working in Poland. Additionally, the research aims to determine the sociodemographic variables and cognitive factors that explain the variability of positive and negative perceptions of COVID-19 pandemic consequences.

This study is timely and interesting and is in line with the theme of this journal. It may be a stimulating paper for the readers of this journal, particularly for those who are interested in learning about the positive aspects of nurses’ perceptions of the aftermath of a pandemic, such as perceived social support, experiencing safety, and a sense of meaning in life.

There are no notable problems with the methods and results. However, certain points should be modified. Therefore, please make the necessary revision by referring to the following.

Overall, the description is too long and gives a verbose impression (Particularly, the abstract and discussion section). Particularly, the discussion section is extremely tedious; thus, I would like you to provide a more concise discussion by holding down the points (by sticking to select important points). This will better convey your main findings to the readers.

Round 2

Reviewer 1 Report

The authors have made all the required modifications. I recommend publishing this paper.